# Organomediated electrochemical fluorosulfonylation of aryl triflates via selective C−O bond cleavage

Xianqiang Kong [1] ✉, Yiyi Chen[1], Xiaohui Chen[1], Cheng Ma[2], Ming Chen [3], Wei Wang[1], Yuan-Qing Xu[4], Shao-Fei Ni [2] ✉ & Zhong-Yan Cao [4] ✉

Although aryl triflates are essential building blocks in organic synthesis, the applications as aryl radical precursors are limited. Herein, we report an organomediated electrochemical strategy for the generation of aryl radicals from aryl triflates, providing a useful method for the synthesis of aryl sulfonyl fluorides from feedstock phenol derivatives under very mild conditions. Mechanistic studies indicate that key to success is to use catalytic amounts of 9, 10-dicyanoanthracene as an organic mediator, enabling to selectively active aryl triflates to form aryl radicals via orbital-symmetry-matching electron transfer, realizing the anticipated C−O bond cleavage by overcoming the competitive S−O bond cleavage. The transition-metal-catalyst-free protocol shows good functional group tolerance, and may overcome the shortages of known methods for aryl sulfonyl fluoride synthesis. Furthermore, this method has been used for the modification and formal synthesis of bioactive molecules or tetraphenylethylene (TPE) derivative with improved quantum yield of fluorescence.

Phenols and its derivatives are ubiquitous in many natural products and bioactive molecules. They can be obtained from industrial Hock process or abundant and renewable biomass such as lignin[1,2]. Therefore, developing transformations based on inexpensive phenols and their derivatives is meaningful[3,4]. Because of the inertness of C−O bonds in phenols, the introduction of an electron-withdrawing group to oxygen is a common activation method. For example, as a typical kind of phenol derivatives, aryl sulfonates[5,6] can serve as electrophiles, participating in versatile transition metal-catalyzed cross-coupling reactions via the oxidation addition of metal catalyst with C−O bonds (two-electron pathway) (Fig. 1a)[7–12]. Besides, recently, excellent examples by using ultraviolet light (254 nm) or the excited state of Pd(0) to promote the formation of very reactive aryl radicals from aryl sulfonates via single electron pathway have been disclosed by Li[13,14] and Gevorgyan[15], respectively. These new activation strategies by connecting the applications of aryl sulfonates with the versatile aryl radical chemistry[16,17] undoubtedly broaden the synthetic scope of phenol derivatives[18–20]. Expanding the potential of aryl sulfonates with a benign activation model for new reaction design is appealing and remains underdeveloped.

On the other hand, driven by the intrinsic advantage of electrosynthesis where an electron could serve as a green reagent to active substrate[21–28] and our work in electrosynthesis[29–33], we envisioned that the reduction of aryl sulfonates at cathode might lead to the generation of aryl radical, similar to aryl halides[34–37]. Furthermore, such an electrochemical activation strategy enables to design of new transformations by employing the advantage of electrosynthesis. However, it is nontrivial to realize such a naive hypothesis as aryl sulfonates have shown different reactivity towards electrolysis than that of aryl halides. This can be attributed to the fact that aryl sulfonates have two different electron-deficient sites (aryl and S atom), and the sulfur atom in aryl sulfonates is the much more electron-deficient one (NBO charge, Fig. 1). Furthermore, upon reduction by chemical reducing reagents[38,39] or direct electrolysis conditions[40,41], several precedent examples have

[1]School of Chemical Engineering and Materials, Changzhou Institute of Technology, No. 666 Liaohe Road, 213032 Changzhou, China. [2]Department of Chemistry, Shantou University, 515063 Shantou, Guangdong, China. [3]Jiangsu Key Laboratory of Advanced Catalytic Materials & Technology, School of Petrochemical Engineering, Changzhou University, 21 Gehu Road, 213164 Changzhou, China. [4]College of Chemistry and Molecular Sciences, Henan University, 475004 Kaifeng, China. ✉e-mail: kongxq@czu.cn; sfni@stu.edu.cn; zycao@henu.edu.cn

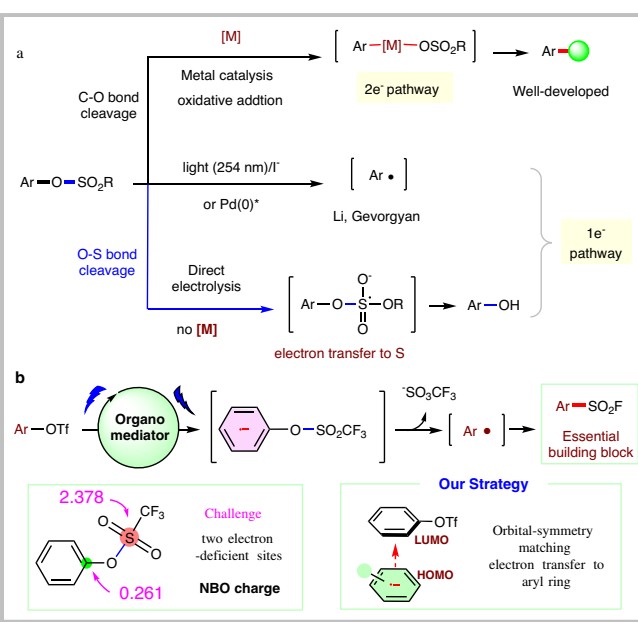

identified that the formation of phenol salts via selective transferring electron to S to cleave S–O bond (Fig. 1a). Therefore, one prerequisite to form aryl radicals from aryl sulfonates is to find a suitable condition which enables to selectively induce the SET process by tuning the transfer of electron from more electron-deficient sulfur atom to the aryl ring. Herein, inspired by recent electrochemical examples which utilize organomediator as an effective electron-transfer mediator[42–55] to activate substrates, we envisioned that by using an orbital-symmetry-matching strategy, whereas electron transfer from the HOMO orbital of a π-ring structure in electron-rich catalytic species to the electron-deficient LUMO orbital of aryl moiety in aryl sulfonates might achieve such selectivity. Indeed, after extensive studies, we found that simple 9,10-dicyanoanthracene could serve as an efficient electron-transfer mediator for the purpose, leading to the formation of key aryl radicals after fragmentation under mild electrochemical conditions. Furthermore, such a strategy enables to achieve the fluorosulfonylation under transition-metal-free conditions with broad functional group tolerance by harnessing the merits of paired electrolysis (Fig. 1b). To the best of our knowledge, although several methods have been reported to prepare the aryl sulfonyl fluorides[56–66], protocols from simple phenol derivatives with broad functional group tolerance has yet to be disclosed[67]. As pointed out by Willis, the use of aryl triflates to realize the fluorosulfonylation via Pd catalysis has failed[68]. Herein, we wish to wish to report our preliminary results.

**Fig. 1 | Selective activation of aryl sulfonates. a** Typical known methods for the selective activation of aryl triflates. **b** Activation of aryl triflates to form aryl radicals enabled by organomediated electrosynthesis.

## Table 1 | Optimization studies[a]

Ar—OTf **1a** (Ar = 4-MeC$_6$H$_4$) + DABSO **2** + KHF$_2$ → Pt(+)|RVC(−), 12 mA, **M-1** (20 mol%), MeCN, $^n$Bu$_4$NClO$_4$, N$_2$, undivided cell, rt, 3 h, "standard conditions" → Ar—SO$_2$F **3a** + Ar—OH **3a'** + (Me—C$_6$H$_4$)$_2$ **3a''** (trace)

| Entry | Alteration | Yield of 3a (3b) (%)[b] |
|---|---|---|
| 1 | None | 75 (0) |
| 2 | **M-2** instead of **M-1** | 41 (35) |
| 3 | **M-3** instead of **M-1** | 49 (31) |
| 4 | **M-4** instead of **M-1** | 0 (82) |
| 5 | **M-5** instead of **M-1** | 0 (82) |
| 6 | **M-6** instead of **M-1** | 0 (88) |
| 7 | without **M-1** | 0 (83) |
| 8 | 25 mol% instead of 20 mol% **M-1** | 74 (0) |
| 9 | 15 mol% instead of 20 mol% **M-1** | 67(0) |
| 10 | $^n$Bu$_4$NPF$_6$, Et$_4$NOTs, LiClO$_4$ instead of $^n$Bu$_4$NClO$_4$ | 70 (0)/52 (0)/ 61 (0) |
| 11 | C(+)|C(−), Pt(+)|C(−), C(+)|Pt(−), RVC(+)|RVC(−), RVC(+)|Pt(−) instead of Pt(+)|RVC(−) | 32 (42)/ 21 (63)/65 (16)/16 (53)/57 (20) |
| 12 | TBAF·H$_2$O, NaPF$_6$, KF instead of KHF$_2$ | 9 (0)/23 (0)/trace (0) |
| 13 | 10 mA or 15 mA instead of 12 mA | 37 (0)/ 56 (8) |
| 14 | No electric current | No reaction |
| 15 | ArOMe, ArOAc, ArOPO(OMe)$_2$, ArBr or ArI instead of **1a** | 0/0/0/0/0 |

**M-1** ($E_{red}$ = -0.95 V)   **M-2** ($E_{red}$ = -1.2 V)   **M-3** ($E_{red}$ = -1.2 V)

**M-4** ($E_{red}$ = -2.6 V)   **M-5** ($E_{red}$ = -2.6 V)   **M-6** ($E_{red}$ = -3.1 V)

[a]Standard conditions: Pt plate (1.0 × 1.0 cm$^2$) anode and RVC cathode was used, $I$ = 12 mA, **1a** (0.20 mmol), **2** (0.15 mmol), KHF$_2$ (0.60 mmol), $^n$Bu$_4$NClO$_4$ (0.05 M), MeCN (4 mL), at room temperature under N$_2$ atmosphere for 3 h.
[b]Isolated yields. DABSO: 1,4-Diazoniabicyclo[2.2.2]octane-1,4-disulphinate. Ag/Ag$^+$ electrode was used as a reference electrode in CH$_3$CN.

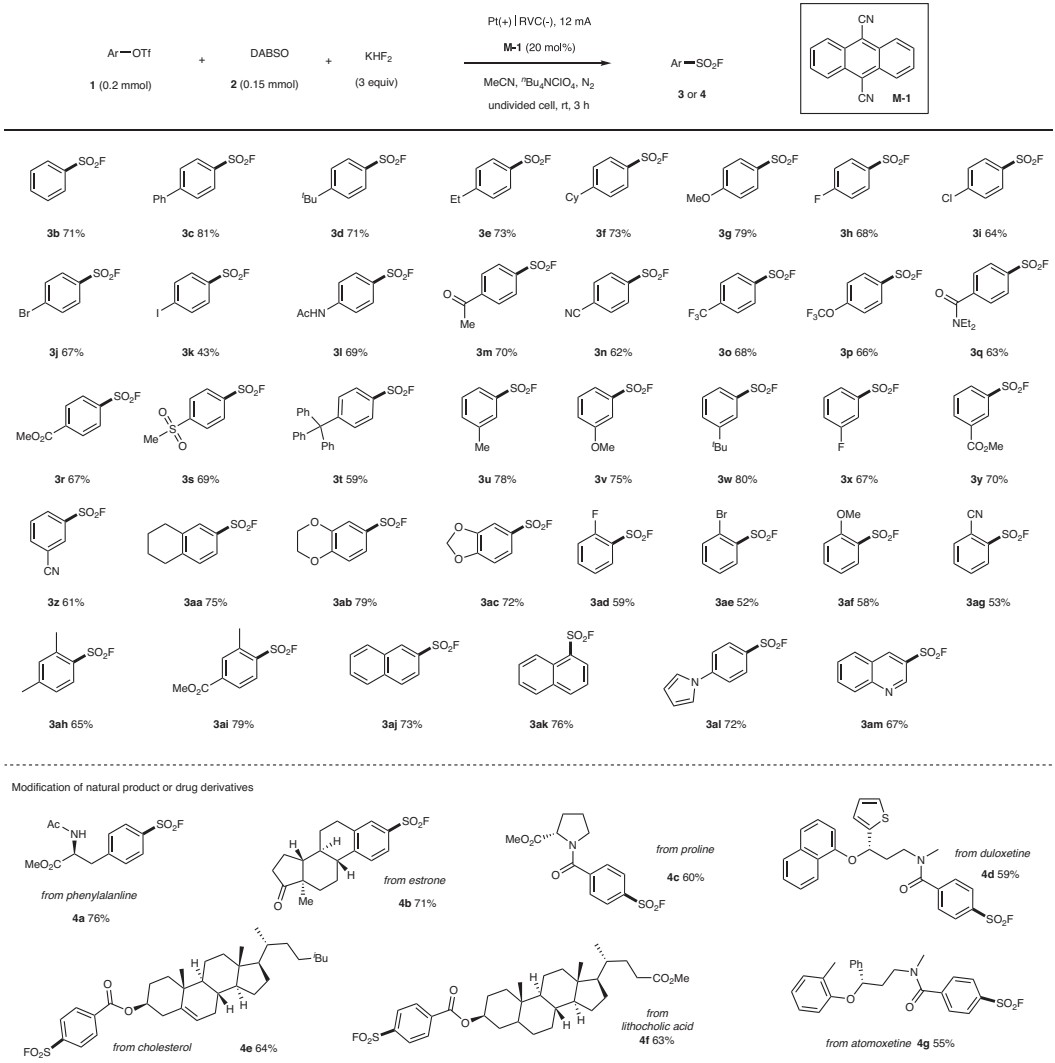

**Fig. 2 | The substrate scope.** For details, please see Supplementary Information (SI).

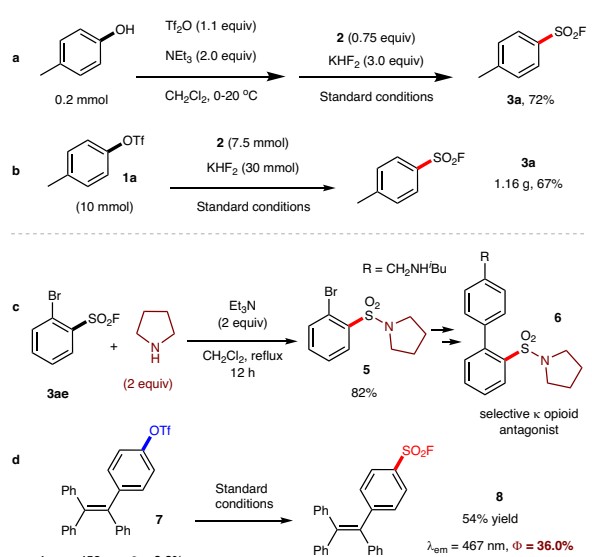

**Fig. 3 | Synthetic applications. a** One-pot procedure from 4-methylphenol. **b** Gram scale reaction. **c** Synthesizing intermediate for bioactive molecule. **d** Synthesizing fluorescent molecule via fluorosulfonylation.

## Results

To optimize the suitable conditions for the designed aryl radical formation from aryl sulfonates by electrochemical method, simple *p*-tolyl trifluoromethanesulfonate **1a** ($E_{red} = -1.0$ V vs. Ag/AgCl) was used as the model substrate, and the fluorosulfonylation had been designed for evaluating the efficiency. As shown in Table 1, the desired fluorosulfonylation could work smoothly, delivering the anticipated **3a** with 75% yield (Faraday efficiency is 74%) under the optimal conditions (20 mol% 9,10-dicyanoanthracene **M-1** as the key organomediator) by overcoming the formation of *p*-cresol **3b** via S–O bond cleavage (entry 1). Noteworthy, a trace amount of 4,4′-dimethyl-1,1′-biphenyl **3c** has also been detected by GC–MS analysis, implying the involvement of 4-methylphenyl radical in this case. During the process, the voltage ranges from 3.49 to 3.1 V, corresponding to a cathodic potential of around −2.4 V vs. Ag/AgCl, and the following five points should be highlighted. (1) As designed, the organomediator **M-1** is essential for the transformation, as the use of other fused polycyclic aromatic compounds or ketones (**M-2**–**M-5**) led to diminished yields along with the observation of sustainable amounts of *p*-cresol **3a′** (entries 2–6). Furthermore, the reactivity is closely associated with the reduction potential of these organomediators, as the formation of *p*-cresol as the major product via S–O bond cleavage was observed in the presence of mediators with a reduction potential less than that of **1a**. As a

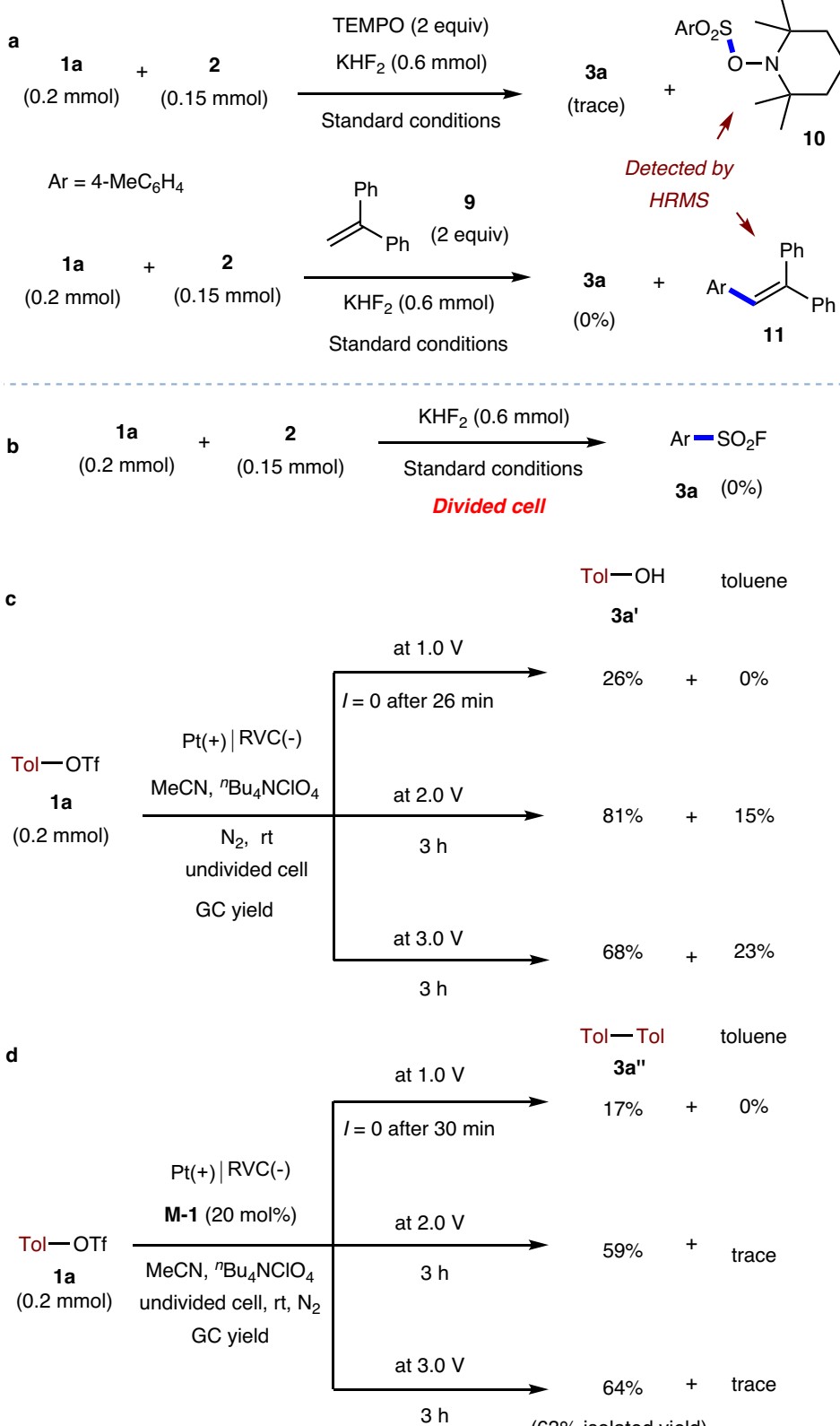

**Fig. 4 | Mechanistic studies. a** Radical trapping experiment. **b** Control experiment in a divided cell. **c** Exhaustive electrolysis of **1a**. **d** Exhaustive electrolysis of **1a** in the presence of **M-1**. TEMPO: 2,2,6,6-Tetramethyl-1-piperidinyloxy. HRMS high-resolution mass spectrometry.

comparison, no **3a** was formed in the absence of organomediator (entry 7). (2) To match the rate of aryl radical formation with the subsequent fluorosulfonylation, 20 mol% loading is optimal (entries 8 and 9). (3) The screen of other electrochemical parameters (electrolyte, electrode, and electric current density) as well as fluoride source implies the essential role of these for good reactivity (entries 10–12). In addition, reducing or increasing the current resulted in lower efficiency (entry 13). (4) Electricity was proven to be essential as no

reaction took place in its absence (entry 14). (5) In comparison, no reaction took place by using 4-MeC$_6$H$_4$OMe, 4-MeC$_6$H$_4$OAc, 4-MeC$_6$H$_4$OPO(OMe)$_2$, 4-MeC$_6$H$_4$Br, or 4-MeC$_6$H$_4$I (the $E_{red}$ of these compounds is −2.6, −2.5, −3.0, −2.6 and −2.5 V, vs. Ag/AgCl, respectively) as aryl source (entry 15).

Having established the optimal condition for the desired fluorosulfonylation, the scope with versatile phenol-derived trifluoromethanesulfonates 1 was carefully evaluated. It turns out that our new strategy shows good functional group tolerance. As shown in Fig. 2, a variety of functional groups such as halides (F, Cl, Br, I), methoxyl, amide, ketone carbonyl, ester, cyano, and sulfonyl were all compatible, giving rise to products 3a–3ac with moderate to good yields (43–81%). In addition, the introduction of substituents at *ortho*- position has little effect on the yield (3ad–3ai). Naphthalene, pyrrole, or quinoline derivatives are suitable, and these products 3aj–3am were isolated with 59–76% yields. Our method can be used for the fluorosulfonylation of natural products or drug derivatives with 55–76% yields (4a–4g). The compatibility with amide, ester, ketone, or thienyl groups highlights the advantage of our protocol.

To demonstrate the synthetic applications, we have conducted the following experiments. (1) The model reaction can be facilely conducted in one pot. As shown in Fig. 3a, the desired 3a could be isolated with 72% from 4-methylphenol after two steps. (2) Our model reaction could be conducted at 10 mmol scale, indicating the practicability of the process (Fig. 3b). (3) In addition, product 3ae can react with tetrahydropyrrole to form sulfonamide 5, a key intermediate for the preparation of selective ĸ opioid antagonist 6[69] (Fig. 3c). (4) The selective fluorosulfonylation from 4-(1,2,2-triphenylvinyl)phenyl trifluoromethanesulfonate 7 enables to deliver 8. More essentially, by using the good electron-withdrawing ability of the FSO$_2$ group, the emission peak of 8 was redshifted to 467 nm from 453 nm of 7, accompanied by an increase of the quantum yield of fluorescence from 3.3% to 36.0% (Fig. 3d).

To shed light on the mechanism, we conducted the following experiments. (1) First, the use of radical scavengers such as 2,2,6,6-tetramethyl-1-piperidinyloxy (TEMPO) or ethene-1,1-diyldibenzene 9 enables to detect the two radical-adducts 11 and 12 by high-resolution mass spectrometry (HRMS), respectively (Fig. 4a). These, together with the detection of 4,4′-dimethyl-1,1′-biphenyl in our model reaction (Table 1), imply the possible involvement of aryl sulfonyl and aryl radicals for our transformation. (2) Second, the model reaction was tested in a divided cell, and no 3a was detected (Fig. 4b), indicating the paired electrosynthesis nature of the reaction. (3) In addition, cyclic voltammetry (CV) studies were conducted to further understand the role of organomediator M-1. As shown in Fig. 5, the reduction peak of 1a and M-1 at −1.0 and −0.95 V vs. Ag/AgCl could be observed (for the full CV and detailed discussions about 1a, please see part 8, Supplementary Information), respectively. Along with the addition of excess amounts of 1a, a decrease in oxidation current and an increase in reduction current of M-1 were detected. (4) To further elucidate the key role of M-1 in mediating the electrochemical reactivity of ArOTf, exhaustive electrolysis experiments have been conducted. As shown in Fig. 4c, while direct electrolysis of 1a at a constant potential of 1.0 V only led to 4-methylphenol 3b, the use of 2.0 and 3.0 V led to the observation of 4-methylphenol 3b and toluene, too. The formation of both 3b and toluene indicates the two different electrochemical pathways for 1a. Nevertheless, after the addition of M-1 (20 mol%), 4-methylphenol 3a′ has been inhibited, with the formation of 4,4′-dimethyl-1,1′-biphenyl 3a′ as the main product (Fig. 4d). These data implied that M-1 could serve as a single-electron transfer mediator for the selective reduction of aryl triflates. The formation of 3a″ with 63% isolated yield also provides a transition-metal-free electrochemical protocol for achieving the reduction cross-coupling between two aryl triflates[40].

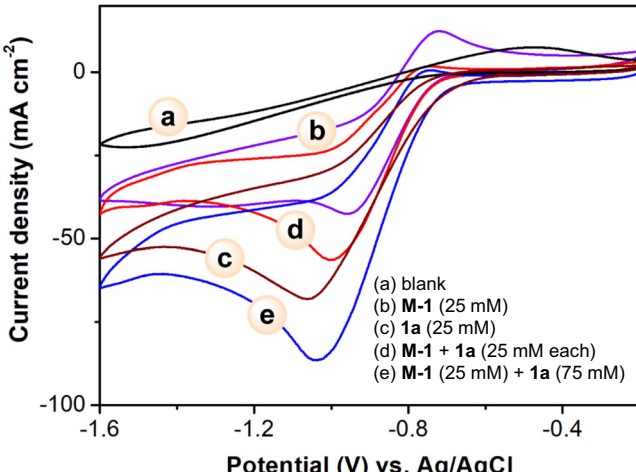

**Fig. 5 | CV experiments.** Experiments were conducted by using glass carbon as the working electrode, Pt plate, and Ag/Ag$^+$ as the counter and reference electrode. Scan rate: 100 mV/s. Solvent: MeCN/"Bu$_4$NClO$_4$ (0.1 M). **a** Background, **b** M-1 (25.0 mM), **c** 1a (25.0 mM), **d** and **e** CVs of M-1 performed in the presence of increasing equivalents of 1a.

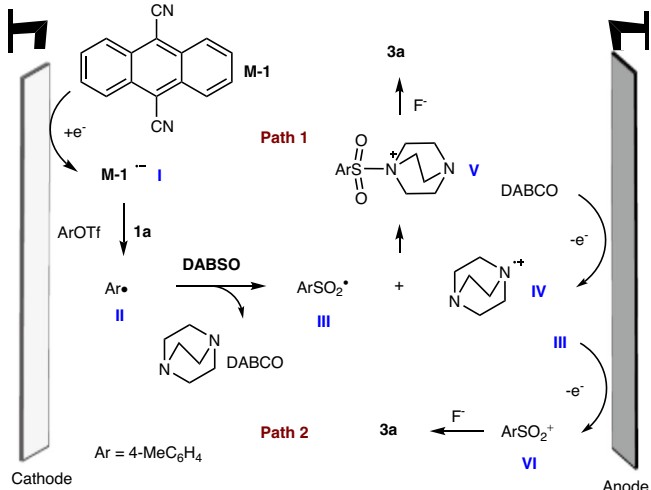

**Fig. 6 | Two plausible mechanisms.** Nucleophilic substitution of 1-(arylsulfonyl) −1,4-diazabicyclo[2.2.2]octan-1-ium IV or sulfonyl cation VI with F$^-$.

Based on the experimental results and literature reports, two plausible mechanisms are proposed for such fluorosulfonylation (using 1a as the model substrate, Fig. 6). At first, M-1 was reduced to a radical anion I at the cathode. After that, intermediate I could selectively reduce 1a to form the key 4-methylphenyl radical II. The subsequent trap by SO$_2$ obtains species III. As postulated by Tlili[70], DABCO will be oxidized to radical cation IV at the anode for path 1. The interaction of III and IV delivers the species V. The subsequent nucleophilic substitution with F$^-$ gives rise to the observed 3a. On the other hand, as proposed by Cheng and Zhou[61], III could be oxidized to form sulfonyl cation VI, and the nucleophilic addition with F$^-$ to form 3a is also possible. Control experiment by using a divided cell led to no 3a (Fig. 4b) also supports these hypotheses.

To give a deep understanding about the selective electron transfer step, additional DFT calculations were conducted. As shown in Fig. 7a, the highest occupied molecular orbital (HOMO) of M-1$^-$ is mainly composed of the π orbital of the anthracene, whose energy level is close to the one of the lowest unoccupied molecular orbital

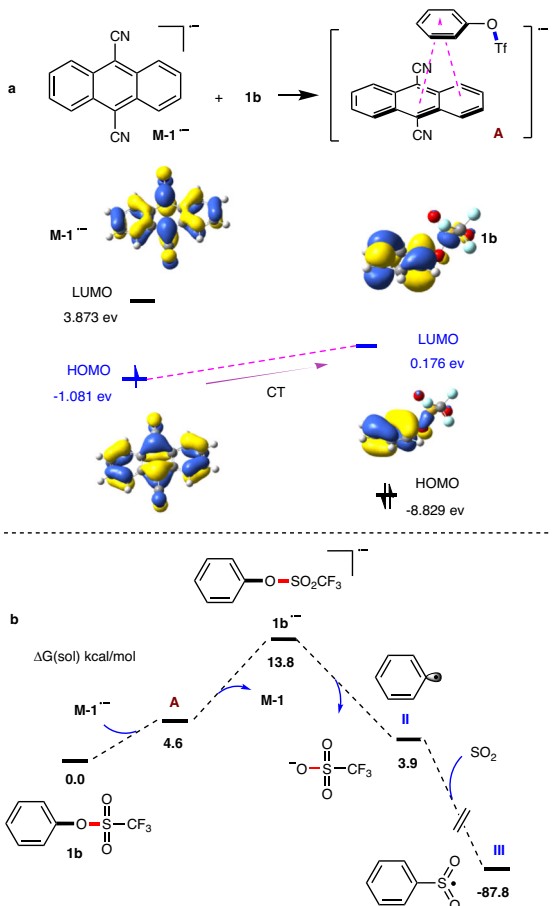

**Fig. 7 | DFT calculations. a** The interaction of HUMO and LUMO orbitals of **M-1**⁻
with **1b**. **b** Computed Gibbs free energy profile for the formation of phenylsulfonyl
radical. HOMO: Highest occupied molecular orbital. LUMO lowest unoccupied
molecular orbital, CT charge transfer.

(LUMO) of **1b**, composed by the π* of the phenyl ring. Charge transfer
(CT) takes place smoothly between the symmetry-matched orbitals
from **M-1**⁻ to **1b**, promoting the formation of adduct **A** (Fig. 7a). After
this, the release of **M-1** from adduct **I** result in radical anion **1b**⁻. DFT
calculation indicates the need for 9.2 kcal/mol energy for this step
(Fig. 7b). Eventually, the facile fragmentation by releasing OTf⁻ group
from **1b**⁻ delivers the key phenyl radical **II**, which will be trapped by
$SO_2$ to deliver radical **III** in a dramatic exothermic way (for details,
please see SI).

## Discussion

In summary, by using 9, 10-dicyanoanthracene as an effective orga-
nomediator, the selective reduction of aryl triflates to aryl radicals via
orbital-symmetry-matching electron transfer has been disclosed, so as
to realize the anticipated C−O bond cleavage by overcoming the
competitive S−O bond cleavage. This enables the development of a
new and practical method for the synthesis of aryl sulfonyl fluorides
from feedstock phenol derivatives under mild electrochemical condi-
tions. The synthetic applications of such fluorosulfonylation from
phenol derivatives have been explored. Further applications of such
strategy for other new reactions are ongoing in our laboratories.

## Methods

### General procedure for the fluorosulfonylation of 1

To the cell was added aryl triflate **1** (0.2 mmol), DABSO (36 mg,
0.15 mmol), $KHF_2$ (46.8 mg, 0.6 mmol), n-$Bu_4NClO_4$ (0.05 M, 68.2 mg),

9, 10-dicyanoanthracene (**M-1**, 9.1 mg, 0.04 mmol), $CH_3CN$ (4 mL). The
tube was installed with a Pt plate (1.0 × 1.0 cm²) as the cathode and
reticulated vitreous carbon (RVC) (1.0 × 1.0 cm²) as the anode. The
mixture was electrolyzed using 12 mA at room temperature under
magnetic stirring. The reaction mixture was poured into ethyl acetate
(40 mL), washed with water two times (10 mL × 2), dried over $Na_2SO_4$,
and concentrated in vacuo. The residue was purified by column
chromatography on silica gel using a mixture of petroleum ether/
EtOAc as eluent to afford the desired pure product **3**, **4**, or **8**.

## Data availability

All the data supporting the findings of this study are available within
the article and its Supplementary Information file. All other data are
available from the corresponding author Xianqiang Kong or Zhong-
Yan Cao.

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

## Acknowledgements

X.Q.K. acknowledges the financial support of the National Natural Science Foundation of China (22372015, 22202021), and the Changzhou Science and Technology Plan Applied Basic Research Project (CJ20210159). Y.C. acknowledges the financial support of the Jiangsu Higher Education Institutions of China (21KJB530013). X.C. acknowledges the financial support of the National Natural Science Foundation of China (22102012, 22272011), Changzhou Science and Technology Plan Applied Basic Research Project (CJ20210129, CZ20220022), and the Jiangsu Higher Education Institutions of China (22KJA150001, 21KJD530003). Z.Y.C. acknowledges the financial support of the National Natural Science Foundation of China (22201062) and Natural Science Foundation of Henan Province (222300420111). S.-F. Ni acknowledges funding from the STU Scientific Research Foundation for Talents (NTF20022).

## Author contributions

Z.C. and X.K. conceived the idea and supervised the whole project. Y.C., X.C., M.C., W.W., and Y-Q.X., carried out the experiments. S.N. and C.M. conducted the DFT calculations. All authors discussed the results, contributed to writing the manuscript, commented on the manuscript, and approved the final version of the manuscript for submission.

## Competing interests

The authors declare no competing interests.
