## [Peer Review File · Nature Communications]

Organomediated Electrochemical Fluorosulfonylation of Aryl Triflates via Selective C–O Bond CleavageReviewers' Comments:

Reviewer #1:

Remarks to the Author:

Professor Kong and coworkers reported an electrochemical synthesis of aryl sulfonyl fluoride from the corresponding aryl triflate in good yield. The key discovery is the application of DCA as the mediator. The conversion of Ar-OTf to Ar-SO₂F has not been reported. From the synthesis application, this work is impressive. On the other hand, the mechanism is still not clear. Several questions should be answered:

1)The redox potential of 4-Me-PhOTf in this work (-1.05 V vs Ag/AgCl, in MeCN from Figure 4) contradict the reported value significantly. (-2.71 V vs SCE, Eur. J. Org. Chem., 1998, 1811-1821, reported in DMF, which is similar to MeCN for CV analysis). The conversion of Ag/AgCl to SCE is approximately -40 mV. (Inorganica Chimica Acta 298 (2000) 97-102)

2)Typically, the reactivity of Ar-OTf is similar to that of Ar-Br in terms of redox potential and oxidative addition in transition metal catalysis. In this work, the reaction of Ar-Br and even Ar-I did not give the product. Please give a explanation on this dramatic contrast.

3)The CV analysis in Figure 4 did not support the proposed mechanism. For the peak potential, the onset potential of 1a (curve C) is almost identical to that of DCA. The stoichiometric and catalytic amount of DCA would be reduced at the cathode in the reaction.

4)In addition, the peak current of DCA is less than that of 1a, suggesting that the reduction of 1a directly would be a major event at the cathode.

These data do not support the mediated reaction pathway in Figure 5.

5)The reaction is conducted under constant current conditions, so the cathode potential might change during the conversion. A reaction at control potential is recommended as the necessary potential for this transformation, which is closely related to the reaction mechanism.

Overall, the mechanism should be re-investigated.

Reviewer #2:

Remarks to the Author:

In this manuscript, Kong, Ni, Cao, and co-workers described an organomediated electrochemical strategy for the synthesis of aryl sulfonyl fluorides from aryl triflates. As described by the authors, the authors use simple 9,10-dicyanoanthracene as the organomediator, which could selectively transfer the electron to the aryl part of aryl triflates, so as to form the key aryl radical. Furthermore, the authors use the advantage of paired electrolysis, which enables to promote the formation of aryl sulfonyl fluorides via the use of cheap KHF₂ as the fluorine source and aryl triflates as the aryl radicals.

In this reviewer's opinion, the most interesting part is that the authors show how the use of organomediator to selectively activate the C-O bond in the presence of thermodynamically more easily reduced position in aryl triflates, in addition to the photo-promoted methods reported by Li and Gevorgyan (strong UV light or expensive Pd metal is necessary).

The use of such a strategy enables to form aryl radical from phenol derivatives at mild reduction potential, compared with other phenol derivatives such as PhOAc, PhOMe, PhOPO(OMe)₂. This reviewer believes such an activation model should promote other new transformations based on phenol derivatives. Furthermore, the synthetic application is interesting, such as the fluorosulfonylation of bioactive molecules, synthesis of a key intermediate for preparing κ opioid antagonist 669, and 4-(1,2,2-triphenylvinyl)benzene-1-sulfonyl fluoride with a good quantum yield of fluorescence.

In view of these, this reviewer recommends the manuscript be published in Nat. Comm. This reviewer has the following questions; please solve them before the final acceptance.

- (1) The authors should specify the reduction potentials of some phenol derivatives such as PhOAc, PhOMe, and PhOPO(OMe)₂, so as to highlight the advantage of such transformation (as low reduction potential will be used in this case).
- (2) The authors proposed the possibility of the reaction mechanism via intermediate V. Is it possible to synthesize such an intermediate, so as to give more evidence to support this hypothesis?
- (3) How about the use of other substrates? Please give a brief description of these results.
- (4) As said before, is such an activation model can be applied to other transformations? Please give at least one example to demonstrate the generality of such a strategy.
- (5) As for the electrodes they used for the reaction, how did the authors clean them after each reaction? Please specify it in the supporting information.
- (6) For some products such as 4c, 4e, and 4f, NMR data show the presence of an atropisomer, please specify them in the SI.

Thanks very much for giving us the opportunity to revise the manuscript (ID: NCOMMS-23-24466). A point-to-point response to all the questions has been provided below. All the changes have been marked **in yellow** in the revised manuscript.

REVIEWER COMMENTS

Reviewer #1 (Remarks to the Author):

Professor Kong and coworkers reported an electrochemical synthesis of aryl sulfonyl fluoride from the corresponding aryl triflate in good yield. The key discovery is the application of DCA as the mediator. The conversion of Ar-OTf to Ar-SO₂F has not been reported. From the synthesis application, this work is impressive.

Our response: We sincerely thank this reviewer's positive comment about our work.

On the other hand, the mechanism is still not clear. Several questions should be answered:

1) The redox potential of 4-Me-PhOTf in this work (-1.05 V vs Ag/AgCl, in MeCN from Figure 4) contradict the reported value significantly. (-2.71 V vs SCE, *Eur. J. Org. Chem.*, 1998, 1811-1821, reported in DMF, which is similar to MeCN for CV analysis). The conversion of Ag/AgCl to SCE is approximately -40 mV. (*Inorganica Chimica Acta* 298 (2000) 97-102)

Our response: We have carefully check this paper (*Eur. J. Org. Chem.* 1998, 1811) and found out that Jutand and co-workers have claimed that the reduction potential of 4-Me-PhOTf is -2.71 V vs SCE about twenty years ago. Very recently, Prof. Chao-Jun Li and co-workers also measured the reduction potential of 4-Me-PhOTf and found out that the value is around -0.9 V vs. Ag/AgCl (*J. Am. Chem. Soc.* 2019, 141, 6755). The data of Li is very closed to ours.

(Copied from the SI of *JACS*, 2019, 141, 6755)

To further identify this, we also measured the CV of 4-Me-PhOTf (scanning from 0 to -3.0 V), and noticed there are actually two reduction peaks at -0.95 and -2.4 V, respectively.

In fact, as pointed out by Li (*J. Am. Chem. Soc.* **2017**, *139*, 8621), there are two electron-deficient sites for ArOTf, as both aryl and S(VI) moiety could accept electron. The transfer of electron to these two different sites could result in two different reduction potentials. Considering that S(IV) moiety is much more electron-deficient than that of aryl moiety, it will be much more easily to be reduced at high reduction potential. In addition, the electrolysis at these two different sites could lead to either phenol or arenes, respectively. **The electrochemical behavior of ArOTf is different from that of ArBr or ArI, as the latter two compounds can be generally reduced to form aryl radical.**

To further prove the above result, we did the **exhaustive electrolysis experiments** by gradually changing the reduction potential. As shown below, while the electrolysis at 1.0 V only led to 4-methylphenol, the use of 2.0 and 3.0 V led to the observation of formation of 4-methylphenol and toluene, too. The experimental result is not only consistent with the electrolysis results of Jutand (*J. Chem. Soc. Chem. Commun.* **1992**, 1729; *J. Org. Chem.* **1997**, *62*, 261), but also further proves the unique electrochemical reactivity of aryl triflates. These identified that the reduction potential of aryl triflate we and Li reported refers to the direct electron transfer to sulfur, while Jutand's data is related to the electron transfer to aryl ring.

To further identify the role of **M-1** for the above electrolysis reaction, we did the control experiments by adding 20 mol% of **M-1**, and it turns out 4-methylphenol has been inhibited, with the formation of 4,4'-dimethyl-1,1'-biphenyl as the main product. These imply that the dramatic role of **M-1** was to be reduced at first at the cathode to its corresponding radical anion. The latter intermediate could selectively transfer the electron, to the aryl moiety of TolOTf, resulting in the formation of aryl radical intermediate, instead of 4-MeC₆H₄OH.

2) Typically, the reactivity of Ar-OTf is similar to that of Ar-Br in terms of redox potential and oxidative addition in transition metal catalysis. In this work, the reaction of Ar-Br and even Ar-I did not give the product. Please give an explanation on this dramatic contrast.

Our response: (1) As for the oxidative addition process of metal with aryl halides or pseudohalides, metal could insert into the C-X bond via a concerted two-electron process. The reactivity of Ar-Br, Ar-I and Ar-OTf is often similar in some cases.

(2) As our response to the first question, the electrochemical behavior between ArOTf and ArBr and ArI is different. For the reduction at the cathode, while ArOTf has two reduction sites and can be reduced to phenols or aryl radicals after the acceptance of electron, both ArBr and ArI has only one reduction site, which lies in their aryl moiety.

(3) As for the reactivity in our electrochemical fluorosulfonylation conditions, the use of ArBr and ArI shows no reactivity comparing with ArOTf (Table 1), which can be attributed

to the following reason.

The reduction potential of 4-MeC₆H₄Br and 4-MeC₆H₄I is -2.6, -2.5 V vs Ag/AgCl. Considering that the cathodic potential we applied for the reaction is around -2.4 V, the organomediator **M-1** should be very easily to be reduced to form intermediate **I**. Indeed, we check the electrolysis reaction of 4-MeC₆H₄Br, 4-MeC₆H₄I and 4-MeC₆H₄OTf in the presence of 20 mol% of **I**, respectively. It turns out these three substrates shows different reactivity.

(i) No change of 4-MeC₆H₄Br was observed in the presence/absence of **M-1**. This is the same as Lin's work (*J. Am. Chem. Soc.* **2020**, *142*, 2087). As pointed out Lin (*J. Am. Chem. Soc.* **2020**, *142*, 2087), simple phenyl bromide cannot be reduced by intermediate **I** because of the poor reduction ability of the latter. In addition, the very negative reduction potential (around -2.6 V) of 4-MeC₆H₄Br makes it difficult to be directly electrolyzed at the cathode.

(ii) As for 4-MeC₆H₄I, while direct electrolysis led to 24% conversion, almost no change of conversion was observed by the addition of 20 mol% of **M-1**. These indicate that 4-MeC₆H₄I cannot be reduced by **I**, either. Instead, 4-MeC₆H₄I can be directly reduced at the cathode because of its reduction potential (around -2.5 V) is higher than that of voltage we used.

(iii) As outlined before, the electrochemical reactivity of 4-MeC₆H₄OTf can be tuned by the organomediator **M-1**, as different reactivity has been observed.

We envisioned that the reason why ArOTf can be reduced by intermediate **I** might be because the good electron-withdrawing and leaving ability of OTf group, making it possible

to acceptor electron from intermediate **I** and formation of aryl radical after fragmentation (*J. Am. Chem. Soc.* **2003**, *125*, 14801; *J. Am. Chem. Soc.* **2020**, *142*, 2087). However, more studies were necessary to elucidate the original difference in the future.

3) The CV analysis in Figure 4 did not support the proposed mechanism. For the peak potential, the onset potential of **1a** (curve C) is almost identical to that of DCA. The stoichiometric and catalytic amount of DCA would be reduced at the cathode in the reaction.

4) In addition, the peak current of DCA is less than that of **1a**, suggesting that the reduction of **1a** directly would be a major event at the cathode.

These data do not support the mediated reaction pathway in Figure 5.

Our response: We sincerely thank this reviewer's good question. It is true that the onset potential of **1a** (curve C) is almost identical to that of DCA when their CV was recorded separately, and the peak current of DCA is less than that of **1a** from Figure 4 (copied below).

(i) First of all, the use of CV to study the interaction between substrate and organomediator is a reliable way in electrochemical synthesis to study the mechanism. For example, to prove that naphthalene could serve as an effective single electron mediator to reduce ArBr, Qiu and co-workers studied the CV between naphthalene and 4-MeOC₆H₄Br (*Angew. Chem. Int. Ed.*, **2022**, *61*, e202210201). In this work, the peak current of naphthalene is lower than that of 4-MeOC₆H₄Br when the CV was measured respectively. However, after the addition of 2.0 equivalents of 4-MeOC₆H₄Br, the CV shows a strong reduction peak at higher reduction potential. It has been used as a very essential evidence to support the authors' mechanism.

(Copied from *ACIE*, **2022**, *61*, e202210201)

In our case, when we conducted the CV analysis, the electrochemical behavior of the reaction mixture (which contains both **1a** and DCA) will be recorded, *not only a single of them*. As shown below, the reaction mixture contains **1a** and DCA with 3:1 ratio shows a very high peak current and onset potential than either of them. This is very similar to the evidence provided by Qiu and co-workers (*Angew. Chem. Int. Ed.*, **2022**, *61*, e202210201) as discussed above. Based on these, these data have been identified as a reliable evidence to

prove the reduction of **1a** by organomediator **M-1**.

Figure 4. (from our manuscript)

(2) In addition, previous control experiments (for answering question 2) by electrolyzing **1a** with or without **M-1** gave different products have identified the role of **M-1** for tuning the electrochemical reactivity of **1a**. Taking together, we envisioned that these data can be used for proving our proposed mechanism.

5) The reaction is conducted under constant current conditions, so the cathode potential might change during the conversion. A reaction at control potential is recommended as the necessary potential for this transformation, which is closely related to the reaction mechanism. Overall, the mechanism should be re-investigated.

Our response: We sincerely thank this reviewer's good suggestion. We measured the cell voltage of the reaction and noticed that the voltage is from 3.49 V to 3.1 V during the whole transformation. In view of this, the model reaction at constant voltage of 0.5, 1.0, 2.0 and 3.0 V has been conducted. It turns out that the reaction took place even at 1.0 V. With the increase in voltage, the yield increases. No 4-MeC₆H₄OH was formed in these cases.

We hope the above explanation could convince the first reviewer to support our proposed mechanism.

Reviewer #2 (Remarks to the Author):

In this manuscript, Kong, Ni, Cao, and co-workers described an organomediated electrochemical strategy for the synthesis of aryl sulfonyl fluorides from aryl triflates.....In

view of these, this reviewer recommends the manuscript be published in Nat. Comm.

Our response: We sincerely thank this reviewer's positive comment about our work.

This reviewer has the following questions; please solve them before the final acceptance.

(1) The authors should specify the reduction potentials of some phenol derivatives such as PhOAc, PhOMe, and PhOPO(OMe)₂, so as to highlight the advantage of such transformation (as low reduction potential will be used in this case).

Our response: These data have been added in the revised manuscript.

(2) The authors proposed the possibility of the reaction mechanism via intermediate V. Is it possible to synthesize such an intermediate, so as to give more evidence to support this hypothesis?

Our response: We thank this reviewer's good suggestion. We have tried to synthesis the following compound according to the literature (*Org. Biomol. Chem.* **2016**, *14*, 8452). However, such salt is very unstable in our hand as the corresponding sulfonyl fluoride can be isolated in the absence of KHF₂. In view of this, it cannot be used as conclusive evidence.

(3) How about the use of other substrates? Please give a brief description of these results.

Our response: We tried the following three types of triflates. However, no desired product could be detected. This information has been added in the revised SI.

(4) As said before, is such an activation model can be applied to other transformations? Please give at least one example to demonstrate the generality of such a strategy.

Our response: Our strategy can be applied for the reductive coupling of ArOTf to give biphenyl. This information has been added in the revised manuscript. Comparing with previous electrochemical work that needs metal such as Pd as catalyst (*J. Chem. Soc., Chem. Commun.* **1992**, 1729), our strategy uses simple catalytic amount of organomediator.

(5) As for the electrodes they used for the reaction, how did the authors clean them after each reaction? Please specify it in the supporting information.

Our response: Thanks for this reviewer's good suggestion. After each reaction, the electrodes were washed with CH₂Cl₂, acetone and EtOH, respectively. This information has been added in the revised supporting information.

(6) For some products such as 4c, 4e, and 4f, NMR data show the presence of an atropisomer, please specify them in the SI.

Our response: Thanks for this reviewer's good suggestion. This information has been added in the revised supporting information.

In addition, we have revised the format based on the editorial policy checklist, and these changes are marked in yellow.

Thank you very much for your help! If you have any other questions, please do not hesitate to contact us as soon as possible.

Yours sincerely

Dr. Xianqiang Kong

Reviewers' Comments:

Reviewer #1:

Remarks to the Author:

The authors revised the manuscript according to both reviewers' comment. The reaction is significantly improved with more results and deep understanding. I would give recommendation for publication. In addition, I suggested to add the discussion on the reduction potential of Ar-OTf including the EJOC paper, recently work of Professor Li, and this work to SI and gave it a index to the discussion in SI in the reference section of Main Article. This would help future researches.

Reviewer #2:

Remarks to the Author:

Now authors have revised the manuscript in the light of the suggestions made by the reviewers and the manuscript shall be accepted for publication.

Thanks very much for giving us the opportunity to revise the manuscript again (ID: NCOMMS-23-24466A). A point-to-point response to all the questions has been provided below. All the changes have been marked **in yellow** in the revised manuscript.

REVIEWER COMMENTS

Reviewer #1 (Remarks to the Author):

The authors revised the manuscript according to both reviewers' comment. The reaction is significantly improved with more results and deep understanding. I would give recommendation for publication. In addition, I suggested to add the discussion on the reduction potential of Ar-OTf including the EIOC paper, recently work of Professor Li, and this work to SI and gave it a index to the discussion in SI in the reference section of Main Article. This would help future researches.

Our response: We sincerely thank this reviewer's positive comment about our work. The discussion on the reduction potential of 4-MeC₆H₄OTf has been added in the part 8, Supplementary Information. In addition, an index to the discussion in SI in the main text has been provided, following the journal's rule.

Reviewer #2 (Remarks to the Author):

Now authors have revised the manuscript in the light of the suggestions made by the reviewers and the manuscript shall be accepted for publication.

Our response: We sincerely thank this reviewer's positive comment about our work again.

In addition, we have revised the format based on the editorial policy checklist, and these changes are marked in yellow.

Thank you very much for your help! If you have any other questions, please do not hesitate to contact us as soon as possible.

Yours sincerely

Dr. Xianqiang Kong